# SEARCH-ON-GRAPH: ITERATIVE INFORMED NAVIGATION FOR LARGE LANGUAGE MODEL REASONING ON KNOWLEDGE GRAPHS

## ABSTRACT

Large language models (LLMs) have demonstrated impressive reasoning abilities yet remain unreliable on knowledge-intensive, multi-hop questions—they miss long-tail facts, hallucinate when uncertain, and their internal knowledge lags behind real-world change. Knowledge graphs (KGs) offer a structured source of relational evidence, but existing KGQA methods face fundamental trade-offs: compiling complete SPARQL queries without knowing available relations proves brittle, retrieving large subgraphs introduces noise, and complex agent frameworks with parallel exploration exponentially expand search spaces. To address these limitations, we propose Search-on-Graph (SoG), a simple yet effective framework that enables LLMs to perform iterative informed graph navigation using a single, carefully designed SEARCH function. Rather than pre-planning paths or retrieving large subgraphs, SoG follows an "observe, think, then navigate" principle: at each step, the LLM examines actual available relations from the current entity before deciding on the next hop. This approach further adapts seamlessly to different KG schemas and handles high-degree nodes through adaptive filtering. Across six KGQA benchmarks spanning Freebase and Wikidata, SoG achieves state-of-the-art performance without fine-tuning. We demonstrate particularly strong gains on Wikidata benchmarks (+16% improvement over previous best methods) alongside consistent improvements on Freebase benchmarks.

## 1 INTRODUCTION

Large language models (LLMs) have demonstrated remarkable capabilities across diverse natural language processing tasks through extensive pre-training on vast text corpora (Brown et al., 2020; Kojima et al., 2022; Wei et al., 2022; Dubey et al., 2024). However, these models face critical limitations when confronted with knowledge-intensive reasoning tasks. They hallucinate plausible-sounding but factually incorrect statements (Tonmoy et al., 2024; Huang et al., 2025), operate with parametric knowledge that becomes rapidly outdated (Liska et al., 2022; Kasai et al., 2023), and lack the specialized domain expertise required for technical fields (Singhal et al., 2023; Kandpal et al., 2023). These limitations are particularly acute in multi-hop reasoning scenarios, where each reasoning step depends on accurate knowledge retrieval and where errors compound across the reasoning chain (Lightman et al., 2023; Ling et al., 2023). Such weaknesses significantly undermine the reliability and trustworthiness of LLMs in real-world applications that demand both factual accuracy and complex reasoning.

To address these challenges, augmenting LLMs with external structured knowledge, specifically knowledge graphs (KGs), has emerged as a promising approach (Sun et al., 2023; Chen et al., 2024; Zhu et al., 2025b). KGs model billions of factual relationships between entities through typed edges. This structured representation supports multi-hop reasoning across diverse domains and allows for efficient updates as knowledge evolves. However, knowledge graph question answering (KGQA) faces significant challenges. KGs like Freebase (1.9 billion triples) (Bollacker et al., 2008) and Wikidata (16 billion constantly-evolving triples) (Vrandečić & Krötzsch, 2014) exemplify the massive scale and dynamic nature of these structures. Additionally, the heterogeneous nature of KG schemas across different repositories makes developing generalizable KGQA methodologies particularly challenging.

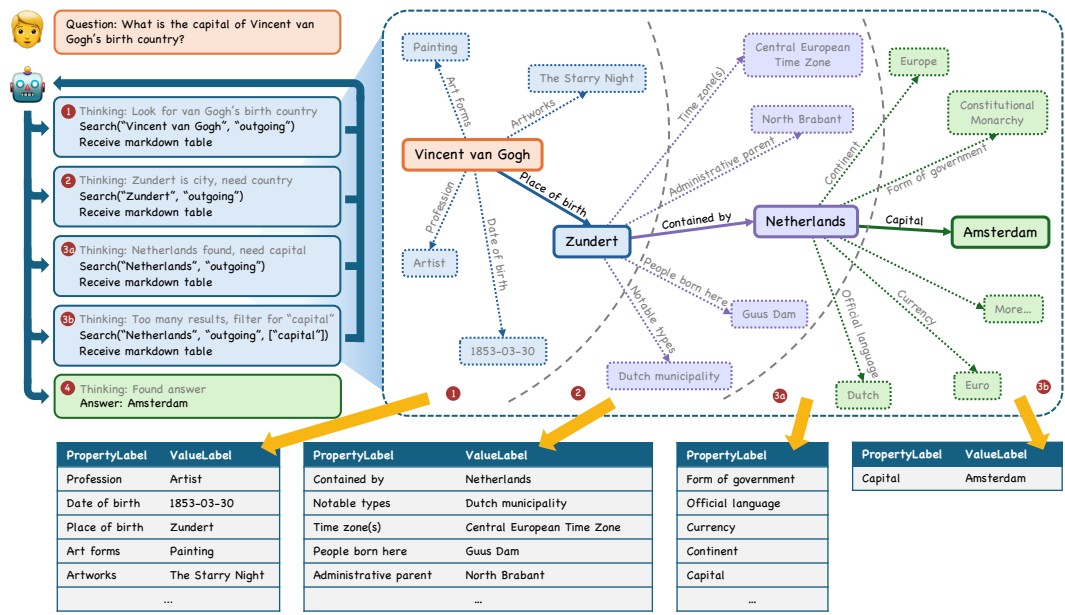

Figure 1: SoG workflow for the query "*What is the capital of Vincent van Gogh's birth country?*" The LLM iteratively navigates the KG, with reasoning and SEARCH calls shown on the left and KG navigation on the right. The path follows Van Gogh → Zundert (place of birth) → Netherlands (contained by) → Amsterdam (capital). Solid boxes indicate selected entities; dotted boxes show unselected retrieved entities. Tables display the markdown output returned by SEARCH, revealing available 1-hop neighbours from each entity.

Existing approaches exhibit inherent limitations. Semantic parsing methods synthesize executable logical forms (SPARQL, S-expressions, etc.) but require extensive schema knowledge and demonstrate limited transferability across different KG architectures (Ye et al., 2021; Yu et al., 2022; Zhang et al., 2023; Luo et al., 2023a; Zhao et al., 2025b; Fang et al., 2024; Zhang et al., 2025; Wulamu et al., 2025). Subgraph retrieval techniques expand entity neighborhoods but frequently extract large, noisy subgraphs that obscure relevant information (Shi et al., 2021; Das et al., 2022; He et al., 2024; Tan et al., 2025; Sun et al., 2018; 2019; Jiang et al., 2022; Zhang et al., 2022). Many employ separate embedding modules for semantic similarity-based subgraph selection (Sun et al., 2018; 2019; Zhang et al., 2022; Ding et al., 2024), yet semantic representations can be misleading—for the query "*What awards did the director of Inception win?*", similarity-based retrievers may include extraneous movie metadata when only the director-award relational pathway is relevant. Recent agentic LLM approaches attempt more targeted path exploration (Sun et al., 2023; Chen et al., 2024; Dong et al., 2024; Luo et al., 2024; Jiang et al., 2024; Wang & Yu, 2025; Cheng et al., 2024; Sui et al., 2024; Wang et al., 2025; Li et al., 2024b) but require complex architectural frameworks, comprehensive upfront planning, or parallel path exploration, thereby increasing computational complexity. Moreover, planning-based approaches introduce failure modes when presumed relations are absent from the actual KG.

In response to these challenges, we propose Search-on-Graph (SoG), a fundamentally simpler methodology where a single LLM orchestrates iterative KG traversal through one carefully engineered SEARCH function executing 1-hop exploration. The key insight is the precedence of observation over speculation—rather than blind path planning or semantic similarity heuristics, the LLM first systematically observes actual available relational connections at each entity, and then formulates informed navigational decisions grounded in question-specific reasoning.

For the query "*What is the capital of Vincent van Gogh's birth country?*" illustrated in Figure 1, the LLM executes iterative SEARCH calls while observing relational options at each hop. From *Van Gogh*, it identifies "*Place of birth*" and navigates to Zundert, subsequently discovers "*Contained by*" to reach Netherlands, and finally selects "*Capital*" to arrive at Amsterdam. This methodology

adapts to heterogeneous schemas—if *Van Gogh* connected directly to *Netherlands* in an alternative KG, the LLM would adopt the shorter path. Our architectural simplicity stems from three deliberate design decisions: (1) an exploration function with compact result formatting that conserves context length, (2) a dynamic filtering mechanism that returns unique relation types when encountering large neighbourhoods, and (3) systematically engineered prompts that demonstrate effective reasoning processes. These seemingly simple design choices prove crucial across diverse KG schemas and question types.

Empirically, SoG delivers strong and consistent gains across six KGQA benchmarks spanning Freebase and Wikidata. Our method achieves state-of-the-art performance on all six datasets, with particularly notable improvements on Wikidata-based benchmarks where we see average gains of over 16% compared to previous best methods. On Freebase datasets, SoG consistently outperforms existing approaches with meaningful improvements across different reasoning complexities. These results demonstrate that our simple, observation-driven design can match or exceed more elaborate architectures while maintaining computational efficiency and broad applicability across different KG schemas.

Our main contributions are as follows:

- We propose a general KGQA framework called Search-on-Graph (SoG) that uses a single LLM with an iterative 1-hop SEARCH function to reliably navigate diverse graph schemas.

- We analyze several design choices (function output format, relation filtering, few-shot examples, and model selection) and show how careful design of these components improves overall performance and efficiency.

- We conduct extensive experiments demonstrating that SoG's simple, plug-and-play design achieves state-of-the-art results across six widely-used KGQA benchmarks.

## 2 RELATED WORK

**Semantic Parsing Methods.**    Semantic parsing techniques transform natural language questions into executable logical forms before KG querying. RNG-KBQA (Ye et al., 2021) enumerates candidate logical forms through KG path searches, then employs ranking and generation models for executable form composition. A different approach is taken by DecAF (Yu et al., 2022), which linearizes KBs into text documents, enabling retrieval-based joint decoding of both logical forms and direct answers. FC-KBQA (Zhang et al., 2023) uses fine-to-coarse composition to address generalization and executability, reformulating KB components into middle-grained knowledge pairs.

LLM-based methods have since emerged to leverage language models' capabilities for logical form generation. ChatKBQA (Luo et al., 2023a) utilizes generate-then-retrieve pipelines, where instruction-tuned LLMs produce candidate logical forms subsequently grounded through phrase-level retrieval. In contrast, CoG (Zhao et al., 2025b) generates fact-aware queries through parametric knowledge output, then corrects hallucinated entities via KG alignment. DARA (Fang et al., 2024) introduces dual mechanisms for high-level task decomposition and low-level task grounding. Meanwhile, Rule-KBQA (Zhang et al., 2025) employs learned rules guiding generation through induction and deduction phases with symbolic agents. HTML (Wulamu et al., 2025) proposes hierarchical multi-task learning with auxiliary tasks for entities, relations, and logical forms.

While these approaches provide interpretable traces and error recovery, they require generating complete logical forms or query plans upfront, demanding extensive schema knowledge and struggling when presumed schema elements are absent. In contrast, navigation based solely on locally available relations adapts to actual KG structures without upfront schema requirements.

**Subgraph Retrieval Methods.**    These approaches involve first retrieving relevant graph portions around topic entities, then proceeds with reasoning over the induced subgraph. GRAFT-Net (Sun et al., 2018) exemplifies early neural approaches by constructing heterogeneous subgraphs that merge KB entities with Wikipedia text, utilizing graph networks with directed propagation for multi-hop inference. PullNet (Sun et al., 2019) employs iterative subgraph expansion using graph CNNs to determine which nodes to "pull" next. TransferNet (Shi et al., 2021) transfers entity scores along activated edges through attention mechanisms while attending to question spans.

More sophisticated retrieval strategies have been proposed to address coverage and noise issues. UniKGQA (Jiang et al., 2022) uses question-relation score propagation along KG edges for unified retrieval-reasoning. SR+NSM (Zhang et al., 2022) employs trainable subgraph retrievers decoupled from reasoning to enable plug-and-play enhancement. CBR-SUBG (Das et al., 2022) dynamically retrieves similar k-NN training queries with structural similarity. G-Retriever (He et al., 2024) formulates subgraph selection as Prize-Collecting Steiner Tree problems, while EPR (Ding et al., 2024) models structural dependencies through atomic adjacency patterns. Paths-over-Graph (Tan et al., 2025) uses multi-hop path expansion with graph reduction and pruning.

These methods face fundamental trade-offs: larger subgraphs boost recall but introduce noise, while smaller ones risk missing critical edges. Furthermore, answer quality is solely dependent on retrieval completeness—key relations filtered during construction cannot be recovered by reasoning modules.

**Agentic LLM Methods.** This paradigm is characterized by interactive KG exploration through LLM agents. Think-on-Graph (Sun et al., 2023) performs iterative beam search maintaining top-$N$ partial paths with pruning. Plan-on-Graph (Chen et al., 2024) decomposes questions into sub-objectives with trajectory memory and reflection mechanisms.

Multi-model approaches are motivated by the need to balance planning and efficiency. EffiQA (Dong et al., 2024) employs LLM global planning combined with lightweight model exploration. KELDaR (Li et al., 2024b) introduces question decomposition trees for atomic KG retrieval. Fi-DeLiS (Sui et al., 2024) combines Path-RAG with deductive beam search, ReKnoS (Wang et al., 2025) uses super-relations enabling bidirectional reasoning, and iQUEST (Wang & Yu, 2025) integrates iterative decomposition with GNNs.

While enabling flexible exploration without complete upfront queries, these approaches often introduce complex multi-component architectures requiring separate modules for planning, memory, and pruning. Most critically, parallel path exploration using beam search exponentially expands the search space, inundating LLMs with extraneous information.

## 3 PRELIMINARIES

### 3.1 KNOWLEDGE GRAPHS

A knowledge graph (KG) $\mathcal{G} = \{(e, r, e') \mid e, e' \in \mathcal{E}, r \in \mathcal{R}\}$ represents structured factual knowledge, where $\mathcal{E}$ and $\mathcal{R}$ denote the entity and relation sets, respectively. Each triple $(e, r, e')$ encodes a factual relationship $r$ between head entity $e$ and tail entity $e'$. Entities are uniquely identified by specific IDs (e.g., `m.07_m2` represents Vincent van Gogh in Freebase) and may possess associated textual labels and semantic types for human interpretation. For any entity $e$, its neighborhood structure comprises both outgoing and incoming relations. We formally define the neighboring relations as $\mathcal{R}_e = \{r \mid (e, r, e') \in \mathcal{G}\} \cup \{r \mid (e', r, e) \in \mathcal{G}\}$, encompassing relations where $e$ serves as either subject or object. This bidirectional connectivity enables flexible traversal during reasoning, allowing navigation in either direction along relational edges.

### 3.2 REASONING PATH

Multi-hop reasoning over KGs requires constructing connected sequences of triples that link topic entities to answer entities. A reasoning path $\mathcal{P}$ of length $k$ from entity $e_0$ to entity $e_k$ is formally defined as:

$$\mathcal{P} = [(e_0, r_1, e_1), (e_1, r_2, e_2), \dots, (e_{k-1}, r_k, e_k)]$$

where each consecutive pair of triples shares an entity, creating a connected traversal through the graph structure. Intermediate entities $e_1, \dots, e_{k-1}$ serve as stepping stones.

Consider the reasoning path illustrated in Figure 1: Vincent van Gogh $\xrightarrow{\text{Place of birth}}$ Zundert $\xrightarrow{\text{Contained by}}$ Netherlands $\xrightarrow{\text{Capital}}$ Amsterdam. This 3-hop path demonstrates how complex questions requiring decompositional reasoning can be decomposed into sequential relational steps, each building upon the previous entity to reach the final answer.

## 3.3 KNOWLEDGE GRAPH QUESTION ANSWERING

Knowledge Graph Question Answering (KGQA) addresses the challenge of answering natural language questions using structured knowledge representations. Given a natural language question $q$, a KG $\mathcal{G}$, and topic entities $\mathcal{T}_q \subseteq \mathcal{E}$ mentioned in $q$, the objective is to identify answer entities $\mathcal{A}_q \subseteq \mathcal{E}$. As per prior work (Luo et al., 2023b; Sun et al., 2023; Chen et al., 2024), we use the gold entity annotations provided in the datasets, where entity mentions in questions are already linked to their KG identifiers, thus bypassing the need for entity linking.

# 4 METHODOLOGY

## 4.1 THE SEARCH FUNCTION

We carefully design a single SEARCH function (Algorithm 1) that enables incremental KG navigation by retrieving the 1-hop neighbours of a target entity in a specified direction. This function serves as the LLM's sole interface for KG exploration via tool calls, accepting three parameters:

- `entity`: The target entity identifier (e.g., `m.07_m2` for Vincent van Gogh)
- `direction`: Either `outgoing` (entity as subject) or `incoming` (entity as object)
- `properties` (optional): Specific properties to filter results for focused exploration

The function returns results in a space-efficient markdown table format, prefixed with a row count that provides the LLM with immediate context about the result size. Each row contains four columns—property ID, property label, value ID, and value label—providing both machine-readable identifiers and human-readable labels. As demonstrated in Figure 1, calling the function to get Vincent van Gogh's outgoing neighbours returns:

```
594 rows:
 property                             | propertyLabel  | value      | valueLabel
 people.person.profession             | Profession     | m.0n1h     | Artist
 visual_art.visual_artist.art_forms   | Art forms      | m.05qdh    | Painting
 people.person.place_of_birth         | Place of birth | m.0vlxv    | Zundert
 people.person.date_of_birth          | Date of birth  | 1853-03-30 | –
 ...                                  | ...            | ...        | ...
```

## 4.2 HANDLING HIGH-DEGREE NODES

KGs often contain high-degree nodes—entities with thousands or millions of connections such as countries, celebrities, or major organizations. Naively retrieving all neighbours of such nodes would overwhelm the LLM's context window and introduce excessive noise. We address this through a two-stage filtering mechanism formalized in Algorithm 1.

---

**Algorithm 1:** ADAPTIVE NEIGHBOURHOOD RETRIEVAL

**Input:** `entity_id`, `direction`, `properties`; thresholds $k, p$
**Output:** 1-hop neighbours of `entity_id` in markdown table format

$R \leftarrow$ GET_ALL_NEIGHBOURS($\texttt{entity\_id}, \texttt{direction}, \texttt{properties}$)

**if** $|R| > k$ **and** `properties` *is empty* **then**
  $U \leftarrow$ EXTRACT_UNIQUE_PROPERTIES($R$)
  **return** FORMAT_AS_TABLE (U)

**if** $|R| > p$ **then**
  $R \leftarrow R[0:p]$

**return** FORMAT_AS_TABLE ($R$)

---

When the function encounters an entity with more than $k$ connected neighbours $R$ without specified properties, our function returns only the unique properties $U$ rather than all neighbour instances. As shown in Figure 1, querying for the *Netherlands* outgoing neighbours returns:

```
property                             propertyLabel
location.country.form_of_government  Form of government
location.country.official_language   Official language
location.country.capital             Capital
...                                  ...
```

This property-only view allows the LLM to first survey available relation types without context overflow. The LLM then makes a targeted second call using the `properties` parameter to retrieve only relevant relations. This transforms high-degree node navigation from an intractable problem into two manageable steps: property discovery followed by selective retrieval.

Even with filtering, results may exceed practical limits. Algorithm 1 shows that when filtered results exceed $p$ triples ($|R| > p$), we truncate to the first $p$ results to ensure the response fits within context limits.

### 4.3 SEARCH-ON-GRAPH PROMPTING

To guide the LLM's navigation strategy, we employ few-shot prompting with navigation exemplars that demonstrate effective KG traversal patterns. For each dataset, we construct five diverse exemplars covering three key aspects:

- **Initial exploration**: strategically making the first SEARCH call based on the question's focus.
- **Iterative traversal**: analyzing retrieved neighbours, selecting a relevant relation, and chaining SEARCH calls to construct reasoning paths.
- **Answer extraction**: recognizing completion conditions and extracting final answers from the reasoning chain.

These exemplars demonstrate to the LLM how to navigate the KG through systematic observation and decision-making. The resulting traces remain fully interpretable as each navigation step is explicitly recorded through tool calls. Appendix A provides the tool definitions, detailed instructions given to the LLM, and representative exemplars for each dataset. Due to space constraints, we include sample exemplars rather than the complete sets used in our experiments.

## 5 EXPERIMENTS

### 5.1 EXPERIMENTAL SETUP

**Datasets and Evaluation Metric.** We evaluate SoG on six KGQA benchmarks spanning two major knowledge graphs, Freebase (Bollacker et al., 2008) and WikiData (Vrandečić & Krötzsch, 2014). For SimpleQuestions (SimpleQA) (Bordes et al., 2015), WebQuestionsSP (WebQSP) (Yih et al., 2016), ComplexWebQuestions (CWQ) (Talmor & Berant, 2018), and GrailQA (Gu et al., 2021), we use Freebase. For QALD-9 (Perevalov et al., 2022) and QALD-10 (Perevalov et al., 2022), we use Wikidata. For SimpleQA and GrailQA, we evaluate on the same 1,000-sample test subset adopted by ToG (Sun et al., 2023) to manage computational costs while enabling direct comparison with prior work. For other datasets, we use the full test sets. As per prior work (Li et al., 2023; Sun et al., 2023; Chen et al., 2024), we report exact match accuracy (Hits@1).

**Models.** We evaluate three off-the-shelf LLMs without fine-tuning: two open-source models—Qwen3-30B-A3B-Thinking-2507 and Qwen3-235B-A22B-Thinking-2507-FP8 (Yang et al., 2025) (abbreviated as Qwen3-30B and Qwen3-235B), and a proprietary model—GPT-4o. SoG is designed as a plug-and-play framework compatible with any LLM that supports tool calling. For GPT-4o, we use the OpenAI API. For Qwen3-30B and Qwen3-235B, we follow the recommended settings (`temperature=0.6, top_p=0.95, top_k=20, min_p=0`).

Table 1: Exact match accuracy (%) of KGQA methods across six benchmarks. Bold and underlined values indicate best and second-best results per dataset, respectively. Datasets are grouped by underlying KG: Freebase (SimpleQA, WebQSP, CWQ, GrailQA) and Wikidata (QALD-9, QALD-10).

| Method | Freebase | | | | Wikidata | |
|---|---|---|---|---|---|---|
| | SimpleQA | WebQSP | CWQ | GrailQA | QALD-9 | QALD-10 |
| *Subgraph Retrieval Methods* | | | | | | |
| GRAFT-Net (Sun et al., 2018) | - | 66.4 | 32.8 | - | - | - |
| PullNet (Sun et al., 2019) | - | 68.1 | 47.2 | - | - | - |
| TransferNet (Shi et al., 2021) | - | 71.4 | 48.6 | - | - | - |
| UniKGQA (Jiang et al., 2022) | - | 77.2 | 51.2 | - | - | - |
| EWEK-QA + GPT-3.5 (Dehghan et al., 2024) | 50.9 | 71.3 | 52.5 | 60.4 | - | - |
| SubgraphRAG + GPT-4o (Li et al., 2024a) | - | 90.9 | 67.5 | - | - | - |
| *LLM Baselines* | | | | | | |
| IO Prompting + Qwen3-30B | 24.8 | 61.1 | 39.0 | 26.7 | 65.1 | 47.2 |
| IO Prompting + Qwen3-235B | 30.3 | 61.1 | 51.0 | 32.3 | 62.7 | 48.7 |
| IO Prompting + GPT-4o | 48.8 | 61.0 | 51.2 | 35.8 | 65.9 | 46.9 |
| *Agentic LLM Methods* | | | | | | |
| Think-on-Graph + GPT-4 (Sun et al., 2023) | 66.7 | 82.6 | 69.5 | 81.4 | - | 54.7 |
| Generate-on-Graph + GPT-4 (Xu et al., 2024) | - | 84.4 | 75.2 | - | - | - |
| Plan-on-Graph + GPT-4 (Chen et al., 2024) | - | 87.3 | 75.0 | 84.7 | - | - |
| Readi + GPT-4 (Cheng et al., 2024) | - | 78.7 | 67.0 | - | - | - |
| Spinach + GPT-4o (Liu et al., 2024) | - | - | - | - | 58.3 | 63.1 |
| FiDeLiS + GPT-4-Turbo (Sui et al., 2024) | - | 84.4 | 71.5 | - | - | - |
| EffiQA + GPT-4 (Dong et al., 2024) | 76.5 | 82.9 | 69.5 | 78.4 | - | 51.4 |
| KELDaR + GPT-3.5-Turbo (Li et al., 2024b) | - | 79.4 | 53.6 | - | - | - |
| ReKnoS + GPT-4o-mini (Wang et al., 2025) | 67.2 | 83.8 | 66.8 | 80.5 | - | - |
| iQUEST + GPT-4o (Wang & Yu, 2025) | - | 88.9 | 73.9 | 73.5 | - | - |
| ORT + GPT-4o (Liu et al., 2025) | - | 87.7 | 65.4 | - | - | - |
| Debate-on-Graph + GPT-4 (Ma et al., 2025) | - | 91.0 | 56.0 | 80.0 | - | - |
| SRP + GPT-4.1-mini (Zhu et al., 2025a) | - | 83.6 | 69.0 | 78.8 | - | - |
| KnowPath + DeepSeek-V3 (Zhao et al., 2025a) | 65.3 | 89.0 | 73.5 | - | - | - |
| **SoG + Qwen3-30B (Ours)** | 86.2 | 88.2 | 70.0 | 81.4 | 81.0 | 77.5 |
| **SoG + Qwen3-235B (Ours)** | **86.4** | 89.3 | **77.1** | 83.9 | **82.5** | **79.8** |
| **SoG + GPT-4o (Ours)** | 84.8 | **91.3** | 75.1 | **86.9** | 79.4 | 74.4 |

**Few-shot Prompting.** For each dataset, we manually construct five few-shot exemplars covering diverse reasoning patterns, including single-hop retrieval, multi-hop traversal, constraint verification, and aggregation. These exemplars are derived from training set questions.

**Baselines and Parameters.** We compare SoG with 23 baselines, grouped into subgraph retrieval methods, LLM baselines, and agentic LLM methods. Semantic parsing methods are excluded due to their reliance on task-specific fine-tuning, which is orthogonal to our training-free paradigm. For all experiments, we set the high-degree threshold $k = 50$ and the maximum result size $p = 1000$, balancing information completeness with context window constraints.

## 5.2 MAIN RESULTS

Table 1 presents the performance of SoG and competing methods across all six benchmarks. Our approach consistently achieves state-of-the-art or highly competitive results using only off-the-shelf LLMs, without any task-specific fine-tuning or retraining. SoG + GPT-4o achieves the highest scores on WebQSP (91.3%) and GrailQA (86.9%), while SoG + Qwen3-235B leads on SimpleQA (86.4%), CWQ (77.1%), QALD-9 (82.5%), and QALD-10 (79.8%). Notably, SoG + GPT-4o outperforms all prior systems on 5 of 6 datasets, trailing only Generate-on-Graph on CWQ by 0.1%. Similarly, SoG + Qwen3-235B also surpasses previous bests on 4 of 6 datasets, with narrow margins of 0.7% and 0.8% behind the previous best on WebQSP and GrailQA, respectively. The improvements over previous best methods range from incremental to substantial. On Freebase datasets, we improve by +0.3% on WebQSP, +1.9% on CWQ, +2.2% on GrailQA, and +9.9% on SimpleQA. The improvements are particularly striking on Wikidata benchmarks, where we achieve double-digit gains: +16.6% on QALD-9 (82.5% vs. 65.9% for IO Prompting) and +16.7% on QALD-10 (79.8% vs. 63.1%).

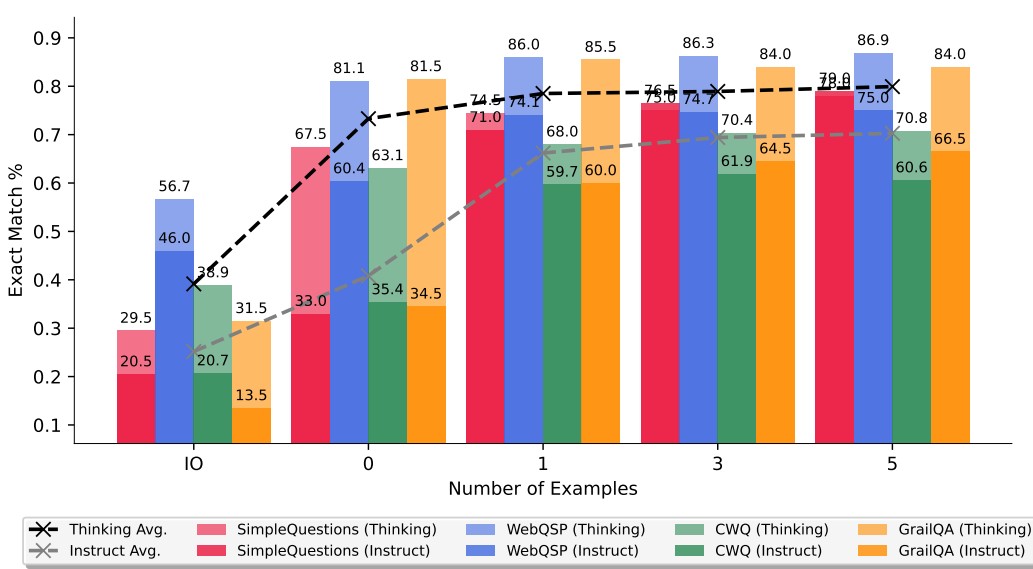

Figure 2: Impact of few-shot exemplar quantity on exact match accuracy (%) across four Freebase datasets (SimpleQA, WebQSP, CWQ, GrailQA) for Qwen3-30B-A3B-Thinking-2507 and Qwen3-30B-A3B-Instruct-2507.

The strong performance across both Freebase (SimpleQA, WebQSP, CWQ, GrailQA) and Wikidata (QALD-9, QALD-10) benchmarks validates our schema-agnostic design. While Freebase uses compound value types for complex relations and Wikidata employs qualifiers, SoG adapts to both structures without modification, confirming that our single function approach generalizes across different KG schemas. Furthermore, SoG is shown to be effective across both single-hop (SimpleQA) and multi-hop (WebQSP, CWQ, GrailQA, QALD-9, QALD-10) datasets. This contrasts with methods like Think-on-Graph, EffiQA, ReKnoS, and KnowPath, which show stronger relative performance only on multi-hop tasks. Our consistent performance across complexity levels likely stems from our focused navigation strategy—by selecting one relation per hop rather than exploring multiple paths in parallel, we avoid the noise accumulation that can overwhelm simpler questions without sacrificing performance on complex reasoning chains.

### 5.3 ABLATION STUDIES AND ANALYSIS

We conduct a series of ablation studies to analyze key design choices in SoG, examining three factors: the impact of few-shot exemplar quantity, reasoning-optimized models versus standard instruction models, and different output formatting on performance. All ablations use 20% of samples from each Freebase-based test set. We evaluate on both Qwen3-30B-A3B-Thinking-2507 and Qwen3-30B-A3B-Instruct-2507.

**Effect of Few-shot Exemplars.** Figure 2 shows the performance of Thinking and Instruct models across varying exemplar quantities. The black and grey dashed lines represent the average exact match accuracy across the four datasets for the Thinking and Instruct models, respectively. Both models show dramatic improvements when transitioning from IO prompting to 0-shot with tool definitions, demonstrating that LLMs can perform structured navigation once they understand the SEARCH function interface. Adding a single navigation exemplar (1-shot) produces another substantial boost across all datasets—from simple single hop to complex multi-hop tasks—confirming that even a single demonstration benefits all complexity levels. Performance plateaus at 3-shot with minimal gains thereafter, indicating that a small set of diverse exemplars sufficiently demonstrates effective navigation strategies.

**Thinking vs. Non-Thinking Models.** The Thinking variant consistently outperforms the Instruct variant across all settings in Figure 2, with the gap most pronounced on multi-hop datasets (WebQSP,

Table 2: Comparison of different output formats on SimpleQA (20% sample) using Qwen3-30B-A3B-Thinking-2507 with 5 exemplars. We report the average number of main interaction tokens, average number of reasoning tokens, average number of total tokens, average number of turns, and exact match (EM) accuracy. "Markdown + Property Filter" denotes our concise format with an additional filtering round, which achieves the best accuracy and efficiency.

| Format | Avg. Main Tok. | Avg. Reason Tok. | Avg. Total Tok. | Avg. Turns | EM |
|---|---|---|---|---|---|
| JSON | 9312.2 | 2735.2 | 12047.3 | 3.06 | 76.5 |
| Markdown | 6028.1 | 1953.7 | 7981.8 | 3.05 | 74.5 |
| Markdown + Property Filter (Ours) | 3715.9 | 1906.6 | 5622.5 | 3.93 | 78.0 |

CWQ, GrailQA) compared to single-hop SimpleQA. This performance difference reveals that model architecture and inherent reasoning capabilities are critical for SoG's effectiveness. The reasoning-optimized model better leverages our iterative observation-decision framework—analyzing available relations and making informed navigation choices based on reasoning rather than question semantics or pattern-matching against exemplars. While both models benefit from additional exemplars, the Thinking variant extracts more value from navigation demonstrations, indicating that SoG's performance ceiling depends on the model's underlying capacity for structured reasoning over KGs.

**Output Format and Filtering.** Table 2 compares the impact of different output formats on performance and efficiency. While the original JSON format that the SPARQL execution returns yields strong accuracy, it uses significantly more tokens than the other two formats. Switching to Markdown format reduces token usage considerably, but slightly impacts accuracy. Our property filtering approach introduces an additional stage: when encountering high-degree nodes, we first retrieve available properties, then make a targeted second call with relevant properties only. Despite requiring additional turns, this strategy achieves the lowest total token usage while simultaneously delivering the highest accuracy. The efficiency gain stems from avoiding redundant information in dense neighborhoods, while the accuracy improvement suggests that focused retrieval helps the LLM identify relevant paths more effectively. These results highlight how careful output design choices critically impacts both computational cost and performance in LLM-based KGQA systems.

## 6 CONCLUSION

We present Search-on-Graph (SoG), a simple yet effective KGQA framework that enables iterative, observation-driven navigation using a single LLM with one carefully designed SEARCH function. This approach achieves state-of-the-art results across six benchmarks spanning Freebase and Wikidata. Our analysis reveals that effective graph navigation depends on three factors: providing LLMs with actual available relations from each entity, using reasoning-optimized models that can leverage navigation demonstrations effectively, and designing output formats that balance information completeness with computational efficiency. Beyond empirical gains, our work demonstrates that many perceived LLM limitations on structured reasoning tasks stem from how we present the problem rather than fundamental model capabilities. By aligning task design with LLM strengths—iterative observation and decision-making over local context—we achieve superior performance without the architectural complexity, specialized modules, or extensive scaffolding that prior work assumed necessary. The simplicity and generality of SoG, requiring no task-specific training and adapting seamlessly across KG schemas, validates this observation-centric approach as a promising direction for LLM-based structured reasoning.

## 7 REPRODUCIBILITY STATEMENT

We provide all materials necessary for reproducibility. Prompts, tool definitions, and few-shot exemplar samples for all datasets are included in Appendix A. Our supplementary materials contain experiment scripts & test data, where `main.py` implements the core processing logic and `prompt.py` contains the system prompt concatenation functions.

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

# A  APPENDIX

## A.1  TOOL DEFINITIONS

Listing 1: Freebase Tool Definition

```
TOOLS_FREEBASE = [
    {
        "type": "function",
        "function": {
            "name": "search",
            "description": (
                "Build and execute a SPARQL query on Freebase that retrieves
                    adjacent properties, property labels,"
                "values, and value labels in the specified direction for a given
                    entity."
            ),
            "parameters": {
                "type": "object",
                "properties": {
                    "entity": {
                        "type": "string",
                        "description": "The entity (e.g., 'm.04yd0fh') whose
                            adjacent relations and entities we want to fetch."
                    },
                    "direction": {
                        "type": "string",
                        "enum": ["incoming", "outgoing"],
                        "description": "Direction of relationship to consider"
                    },
                    "properties_to_filter_for": {
                        "type": "array",
                        "items": {"type": "string"},
                        "description": "Optional list of specific properties to
                            filter by (e.g., ['people.person.place_of_birth', '
                            people.person.nationality'])."
                    }
                },
                "required": ["question", "entity", "direction"],
                "additionalProperties": False
            },

        }
    },
]
```

Listing 2: Wikidata Tool Definition

```
TOOLS_WIKIDATA = [
    {
        "type": "function",
        "function": {
            "name": "search",
            "description": (
                "Build and execute a SPARQL query on Wikidata that retrieves
                    adjacent properties, property labels,"
                "values, and value labels in the specified direction for a given
                    entity."
            ),
            "parameters": {
                "type": "object",
                "properties": {
                    "entity": {
                        "type": "string",
                        "description": "The entity (e.g., 'wd:Q187805') whose
                            adjacent relations and entities we want to fetch.",
                    },
                    "direction": {
                        "type": "string",
                        "enum": ["incoming", "outgoing"],
                        "description": "Direction of relationship to consider",
```

```
21                    },
22                    "properties_to_filter_for": {
23                        "type": "array",
24                        "items": {"type": "string"},
25                        "description": "Optional list of specific properties to
                                filter by (e.g., ['people.person.place_of_birth', '
                                people.person.nationality'])."
26                    }
27                },
28                "required": ["question", "entity", "direction"],
29                "additionalProperties": False,
30            },
31
32        },
33    },
34 ]
```

## A.2   Tool Instruction Prompt

### Listing 3: Tool Instructions

```
You are a knowledgeable question-answering agent specializing in
    knowledge-graph question answering. You will receive a question and
    may call a tool to navigate the knowledge graph, collect information
    , and then formulate an answer.

You may call the tool search(entity, direction) to retrieve adjacent
    relations and 1-hop neighbouring entities to the entity given in the
     input. Additionally, direction must be incoming or outgoing.

When you want to call the tool:
  - Always follow the CORRECT format whenever you want to make a tool
      call.
  - Continue making tool calls until you arrive at a final textual
      answer. Then, and only then, stop making tool calls and provide
      your final answer in 'content'.

Furthermore,
  - Sometimes the 'search' tool returns an entity ID ('value') without
      a corresponding entity name ('valueLabel'). If that occurs,
      continue making the correct tool calls using the entity ID ('value
      ') alone, if necessary, until you find the information needed to
      answer the question. Relevant details may appear in subsequent
      tool calls.
  - Whenever 'search' returns multiple entities for a single relevant
      relation, you must examine every single one of those entities,
      even if there are tens or hundreds. Do not skip any; each could be
       relevant to the question.
  - If the question happens to be a 'when' question, you must provide
      the final answer with the value of the entity as given (i.e., in
      the format {1889-04-20} or {1889-04-20-08:00}) from the results of
       'search'.
  - If searching from one direction does not yield information that
      seems relevant to the question, you may try searching from the
      other direction (e.g., "incoming" instead of "outgoing", or "
      outgoing" instead of "incoming") of the starting entity if you
      think it makes sense to try.
  - In your final answer, you must 1) write 'Final answer:' immediately
       before providing your final answer, 2) enclose the answer entity
      (or entities) in curly braces, and 3) use each entity name exactly
       as returned by the 'search' tool. For example, if the tool's
      output shows "English Language", you must produce {English
      Language} (keeping the exact phrase) rather than shortening it to
      "English.".
```

```
– If you cannot gather enough information using the tools to answer
    the question, rely on the information you already have, your
    reasoning abilities, and your own knowledge to produce the best
    possible answer(s).
```

## A.3  SAMPLE EXEMPLARS

Listing 4: SimpleQA Sample Exemplar

```
Question: where did the continental celtic languages originate? {'
    Continental Celtic languages': 'm.06v3q8'}
Outgoing relations from m.06v3q8 (Continental Celtic languages)
Tool Call: [{"id": "some_unique_id", "type": "function", "function": {"
    name": "search","arguments": {"entity": "m.06v3q8", "direction": "
    outgoing"}}}]
Suppose it returns:
property|propertyLabel|value|valueLabel
--|--|--|--
language.language_family.member_of_language_families|member of language
     families|m.01sd8|Celtic languages
language.language_family.geographic_distribution|geographic
    distribution|m.02j9z|Europe
kg.object_profile.prominent_type||language.language_family|Language
    Family
We see there is a language.language_family.geographic_distribution (
    geographic distribution) property that points to the object m.02j9z
    (Europe).
Final answer: The Continental Celtic languages originated in {Europe}.
```

Listing 5: WebQSP Sample Exemplar

```
Question: what is cher 's son 's name {'Cher': 'm.01vtj38', 'Male': 'm
    .05zppz'}
'm.01vtj38' ('Cher') is the topic entity of the question. Look for
    outgoing edges from 'm.01vtj38' and look for relations and entities
    related to Cher's son. Any linked object entities are potential
    candidates.
Tool Call: [{"id": "some_unique_id", "type": "function", "function": {"
    name": "search","arguments": {"entity": "m.01vtj38", "direction": "
    outgoing"}}}]
Suppose it returns:
property|propertyLabel|value|valueLabel
--|--|--|--
base.schemastaging.context_name.official_name|Official name|Cherilyn
    Sarkisian|
people.person.children|Children|m.01w4bt1|Elijah Blue Allman
people.person.children|Children|m.0br66|Chaz Bono
people.person.parents|Parents|m.0kmhsk2|Gilbert Hartmann LaPiere
We see there is the people.person.children (Children) property that
    points to entities m.01w4bt1 (Elijah Blue Allman) and m.0br66 (Chaz
    Bono).
To specifically look for son(s) of Cher, we check the gender of each of
     those entities
Tool Call: [{"id": "some_unique_id", "type": "function", "function": {"
    name": "search","arguments": {"entity": "m.01w4bt1", "direction": "
    outgoing"}}}]
Suppose it returns:
property|propertyLabel|value|valueLabel
--|--|--|--
people.person.parents|Parents|m.01vtj38|Cher
people.person.gender|Gender|m.05zppz|Male
people.person.sibling_s|Siblings|m.0w4gdrb|
people.person.sibling_s|Siblings|m.0vvfthw|
```

```
We see that Elijah Blue Allman's gender is Male.
Tool Call: [{"id": "some_unique_id", "type": "function", "function": {"
    name": "search","arguments": {"entity": "m.0br66", "direction": "
    outgoing"}}}]
Suppose it returns:
property|propertyLabel|value|valueLabel
--|--|--|--
people.person.parents|Parents|m.01vtj38|Cher
people.person.sibling_s|Siblings|m.0w4gdrb|
people.person.gender|Gender|m.05zppz|Male
We see that Chaz Bono's gender is Male.
Final answer: Cher's sons' names are {Elijah Blue Allman} and {Chaz
    Bono}.
```

Listing 6: CWQ Sample Exemplar

```
Question: What structure build on June 5, 2007 is Charlotte, North
    Carolina known for? {'Charlotte': 'm.0fsb8'}
'm.0fsb8' ('Charlotte') is the topic entity of the question. Look for
    outgoing edges from 'm.0fsb8' and look for relations and entities
    related to structures built in Charlotte. Any linked object entities
     are potential candidates.
Tool Call: [{"id": "some_unique_id", "type": "function", "function": {"
    name": "search","arguments": {"entity": "m.0fsb8", "direction": "
    outgoing"}}}]
Suppose it returns:
property|propertyLabel|value|valueLabel
--|--|--|--
common.topic.topical_webpage|Topical webpage|http://www.charmeck.org/|
travel.travel_destination.tourist_attractions|Tourist attractions|m.09
    k6h_2|Bechtler Museum of Modern Art
travel.travel_destination.tourist_attractions|Tourist attractions|m.02
    vnhrq|Billy Graham Library
travel.travel_destination.tourist_attractions|Tourist attractions|m.05
    g_v0l|Bojangles' Coliseum
travel.travel_destination.tourist_attractions|Tourist attractions|m.0
    cq5c0|Carolinas Aviation Museum
We see the property travel.travel_destination.tourist_attractions (
    Tourist attractions), which points to m.09k6h_2 (Bechtler Museum of
    Modern Art), m.02vnhrq (Billy Graham Library), and m.05g_v0l (
    Bojangles' Coliseum). These are all tourist attractions in Charlotte
    , North Carolina.
Get outgoing relations and entities from each candidate entity to find
    information on the date that it was built:
Tool Call: [{"id": "some_unique_id", "type": "function", "function": {"
    name": "search","arguments": {"entity": "m.09k6h_2", "direction": "
    outgoing"}}}]
Suppose it returns:
property|propertyLabel|value|valueLabel
--|--|--|--
type.object.type|Type|common.topic|Topic
type.object.type|Type|architecture.building|Building
type.object.type|Type|architecture.structure|Structure
There is no property that indicates the build date of m.09k6h_2 (
    Bechtler Museum of Modern Art).
Tool Call: [{"id": "some_unique_id", "type": "function", "function": {"
    name": "search","arguments": {"entity": "m.02vnhrq", "direction": "
    outgoing"}}}]
Suppose it returns:
property|propertyLabel|value|valueLabel
--|--|--|--
common.topic.notable_types|Notable types|m.01y2hbz|Museum
architecture.structure.opened|Opened|2007-06-05-08:00|
type.object.type|Type|base.type_ontology.abstract|Abstract
```

```
We see that there is the property architecture.structure.opened (Opened
    ), which points to the date 2007-06-05-08:00. This indicates an
    opening date of 2007-06-05 (June 5, 2007), which matches our target
    date.
Tool Call: [{"id": "some_unique_id", "type": "function", "function": {"
    name": "search","arguments": {"entity": "m.05g_v0l", "direction": "
    outgoing"}}}]
Suppose it returns:
property|propertyLabel|value|valueLabel
--|--|--|--
architecture.structure.opened|Opened|1955-08:00|
common.topic.social_media_presence|Social media presence|http://www.
    facebook.com/pages/Bojangles-Coliseum/196122978761|
common.topic.social_media_presence|Social media presence|https://
    twitter.com/BojanglesCol|
We see that there is the property architecture.structure.opened (Opened
    ), which points to the date 1955-08:00. This indicates an opening
    date of 1955 at 8am, which does not match our target date of June 5,
     2007.
Final answer: Charlotte, North Carolina is known for the structure {
    Billy Graham Library} that is built on June 5, 2007.
```

Listing 7: GrailQA Sample Exemplar

```
Question: what is the language regulator of basque? {'basque': 'm.017k6
    '}
'm.017k6' ('basque') is the topic entity of the question. Look for
    incoming edges from 'm.017k6' and look for relations and entities
    related to language regulators of Basque. Any linked object entities
     are potential candidates.
Tool Call: [{"id": "some_unique_id", "type": "function", "function": {"
    name": "search","arguments": {"entity": "m.017k6", "direction": "
    incoming"}}}]
Suppose it returns:
property|propertyLabel|value|valueLabel
--|--|--|--
base.rosetta.rosetta_document.refers_to|Refers To|m.05tr3c6|Basque
    Numbers
language.language_regulator.language|Language|m.057xsn|Euskaltzaindia
type.type.instance|Instance|language.languoid|
We see the property language.language_regulator.language (Language),
    which points to m.057xsn (Euskaltzaindia). This may be the language
    regulator of Basque. Let's double check by calling the tool to look
    at its outgoing edges.
Tool Call: [{"id": "some_unique_id", "type": "function", "function": {"
    name": "search","arguments": {"entity": "m.057xsn", "direction": "
    outgoing"}}}]
Suppose it returns:
property|propertyLabel|value|valueLabel
--|--|--|--
type.object.type|Type|common.topic|Topic
type.object.type|Type|base.type_ontology.agent|Agent
type.object.type|Type|language.language_regulator|Language Regulator
We see that there is the property type.object.type (Type), which points
     to language.language_regulator (Language Regulator). This confirms
    that m.057xsn (Euskaltzaindia) is indeed a language regulator.
Final answer: The language regulator of Basque is {Euskaltzaindia}.
```

Listing 8: QALD Sample Exemplar

```
Question: In which country does the Ganges start? {'Ganges': 'wd:Q5089
    '}
wd:Q5089' ('Ganges') is the topic entity of the question. Look for
    outgoing edges from 'wd:Q5089' and look for relations and entities
```

```
        related to which country the Ganges starts in. Any linked subjects
        are potential candidates.
Tool Call:[{"id": "some_unique_id", "type": "function", "function": {"
    name": "search","arguments": {"entity": "wd:Q5089", "direction": "
    outgoing"}}}]
Suppose it returns:
property|propertyLabel|value|valueLabel
--|--|--|--
wdt:P885|origin of the watercourse|wd:Q691557|Gangotri Glacier
wdt:P974|tributary|wd:Q3635865|Punpun River
wdt:P30|continent|wd:Q48|Asia
We see the property wdt:P885 (origin of the watercourse) that links to
    the entity wd:Q691557 (Gangotri Glacier).
Look at each candidate entity's outgoing relations for information
    regarding its country
Tool Call: [{"id": "some_unique_id", "type": "function", "function": {"
    name": "search","arguments": {"entity": "wd:Q691557", "direction": "
    outgoing"}}}]
Suppose it returns:
property|propertyLabel|value|valueLabel
--|--|--|--
wdt:P4552|mountain range|wd:Q3777888|Gangotri Group
wdt:P31|instance of|wd:Q35666|glacier
wdt:P17|country|wd:Q668|India
We see the property wdt:P17 (country) that links to the entity wd:Q668
    (India).
Final Answer: The Ganges starts in {India}.
```

