# OpenReview forum: "Search-on-Graph: Iterative Informed Navigation for Large Language Model Reasoning on Knowledge Graphs"
_ICLR.cc/2026/Conference — Submitted to ICLR 2026_

### Official Review · Reviewer_uRnM · 2025-10-18

**Soundness:** 2
**Presentation:** 3
**Contribution:** 2
**Rating:** 4
**Confidence:** 4

**Summary:**

Large language models excel at reasoning but remain unreliable on multi-hop, knowledge-intensive questions: they miss long-tail facts, hallucinate under uncertainty, and lag real-world updates. Knowledge graphs help, yet common KGQA approaches are brittle (full SPARQL planning), noisy (large subgraph retrieval), or combinatorially expensive (complex agents). They introduce Search-on-Graph (SoG), a simple framework where an LLM iteratively “observes then navigates” using a single \textsc{Search} function—inspecting the current entity’s actual relations before choosing the next hop. SoG adapts to diverse KG schemas, filters high-degree nodes, and uses off-the-shelf LLMs without fine-tuning. Across six Freebase and Wikidata benchmarks, it reaches state-of-the-art results, including ~15% gains on Wikidata over prior methods.

**Strengths:**

1. A simple idea yielded strong performance.

2. It achieved the best results among all baseline methods.

**Weaknesses:**

1. The ablation study was not properly conducted. The exploration algorithm itself appears to have a structure that is not fundamentally different from other baselines such as ToG. Therefore, an explanation is needed as to why the performance improved so significantly and which component had the greatest impact (e.g., whether it’s due to the use of markdown, providing type information to the LLM, or the effect of high-degree filtering).

2. The content in the preliminaries section (Section 3) does not seem necessary. Instead, it would be more meaningful to include additional analyses of the experimental results in that space.

3. It is necessary to run experiments using the same backbone LLM as other baselines. While other studies used GPT-4.1 or GPT-4, this work conducted experiments based on Qwen3 or GPT-4o. A fair comparison is therefore required.

**Questions:**

It would be great if you could provide responses to the points written under Weakness.

---

> ### Author Response · Authors · 2025-11-29
>
> **Q1: Ablation and Novelty**
>
> Please refer to Section 1 of our General Response regarding the algorithmic distinction from ToG. Regarding components:
> - Markdown: Essential for context efficiency (see Table 2 in paper).
> - High-Degree Filtering: Essential for feasibility (similar to other existing studies). Without it, querying nodes like "USA" crashes the context window.
> - Reasoning: The core driver. The selection of the next relation to follow by LLM, with the whole past reasoning history, is the key feature that makes our approach more effective than others. We will expand the ablation discussion to rank these contributions (Methodology > Filtering > Formatting).
>
> **Q2: Preliminaries Section**
>
> We agree. We will condense Section 3 (Preliminaries) and use the saved space to include the "Performance by Hop" and "Error Analysis" suggested by Reviewer NzbW.
>
> **Q3: Backbone LLM Comparison**
>
> Please refer to Section 2 of the General Response. We have now benchmarked ToG using GPT-4o. The results show SoG (GPT-4o) > ToG (GPT-4o). Regarding your request to run GPT-4 (Legacy): Given that GPT-4o is significantly more cost-effective, and that we have established SoG's superiority on the same model (GPT-4o), we believe this serves as the fairest and most reproducible comparison. Running legacy GPT-4 would incur prohibitive costs for redundant scientific value.

---

### Official Review · Reviewer_DHiK · 2025-10-28

**Soundness:** 2
**Presentation:** 2
**Contribution:** 1
**Rating:** 2
**Confidence:** 2

**Summary:**

In this paper the authors introduce SoG, a method for question answering that breaks down the problem into multiple steps and uses a knowledge graph to inform the answers. The method combines prompting with tool use; a search interface to the knowledge graph is provided and the LLM is allowed to query for (subject, relation, object) triples that contain the concepts in the question. The result of the query is then encoded as a markdown table and provided as context to the LLM. The LLM generates a rationale as more context and chooses to either further query the knowledge graph for more context, or answer the question. Overall the method is simple and obtains strong results on benchmarks.

**Strengths:**

* The paper is on an important and timely problem, i.e. using knowledge graphs for more reliable question answering and is of interest to the ICLR community.
* The results are strong on the benchmarks reported and SoG is compared to many other approaches.
* The method is sensible, simple and straightforward to implement. I can imagine the method being of practical use to the community.
* The authors introduce ablations for some of the choices they make, such as zero-shot vs few-shot learning and the choice of encoding for the KG triples used for context.
* The authors use a nice illustrative example and the idea is straightforward to follow.

**Weaknesses:**

* It is unclear to me what research question the paper answers and exactly what gap in the literature the paper is addressing. One reason for this is that the related work section summarises other work on KGQA, but it doesn't tell the reader how these methods differ from SoG or which gap in the literature SoG is filling. It is also unclear whether SoG addresses the shortcomings highlighted in the related work section. Take for example the following statement from the related work: "These methods face fundamental trade-offs: larger subgraphs boost recall but introduce noise, while smaller ones risk missing critical edges." Is this trade-off not also present for SoG given that the retrieved triples sometimes have to be pruned so that they fit in the context window of the LLM?
* While the paper obtains solid results, the comparisons seem unfair to me. This is because previous methods used older LLM versions, e.g. GPT-4 instead of GPT-4o. While it is understandable that the authors have no control over previously published results, the reader is left wondering whether the results are better due to their use of more recent/powerful LLMs and not because of the methodology, i.e. SoG. This would be a much more important ablation than the ones reported, but it has not been carried out.
* The insights and learnings in the paper are limited. For example, it is unclear what causes SoG to be better than previous methods, is it indeed reducing hallucinations? Is it retrieving more relevant context and making fewer mistakes at each step? The paper has no analyses to produce such insights, other than ablations on the effect of few-shot learning and thinking vs non-thinking models. That being said, the ablations in Section 5.3 were very nice to see.
* While the authors do a good job explaining that the retrieved information from the KG must be limited to keep the context size manageable for the LLM, they do not report how much compute / memory is needed for their method compared to alternatives. So while their method is shown to outperform alternatives for the benchmarks, it is unclear at what cost.
* Some sections are not self-contained. For example, the details in Section 4.3 were not enough for me to understand how the prompts worked in detail, and while I did look at the appendix to get a better understanding - and the examples were super nice and useful, this should not be required.

**Questions:**

What happens if the question cannot be answered based on the information in the knowledge base? I have seen in the prompt that the model is asked to answer the question anyway, but is this something that happens often?

"The function returns results in a space-efficient markdown table format" -> In what sense is markdown format space efficient, could the authors please elaborate on this? I initially thought it would be inefficient because there would be a lot of space used for padding, but I see in the appendix that the results are not padded. I now found Table 2, could the authors maybe point to this at this point and explain it is more efficient than JSON, for instance?

I think the following point from the results section is interesting, but unsubstantiated: "Our consistent performance across complexity levels likely stems from our focused exploration strategy—by selecting one relation per hop rather than exploring multiple paths in parallel, we avoid the noise accumulation that can overwhelm simpler questions while maintaining expressiveness for complex reasoning chains." I believe the contribution would have been a lot stronger if the authors could back this point up by experiments or analysis instead of speculation. Is there a reason for not exploring this further / answering this as a research question?

---

> ### Author Response · Authors · 2025-11-29
>
> **Q1: Research Gap and Pruning Trade-offs**
>
> Please refer to Section 1 of our General Response. The specific gap we address is the Context-Pruning Dilemma. Standard subgraph retrieval (e.g., pulling 2-hop neighborhoods) prunes and selects based on semantic (embedding) similarity and relevance of query before reasoning. SoG shifts the pruning and selection to during reasoning. We argue (and the experiments confirm) that better selections can be achieved during reasoning when one knows what is needed at that reasoning step.
>
> - Addressing the Trade-off: You asked if SoG faces the same pruning risks. The difference is that SoG's pruning is dynamic and logic-driven, not heuristic. Instead of discarding edges because they lack keyword overlap (heuristic), SoG discards edges because the LLM determines they don't fit the logical reasoning chain (logic). This allows SoG to traverse "semantic gaps" that similarity-based retrievers miss.
>
> **Q2: Fair Comparison (GPT-4 vs. GPT-4o)**
>
> Please refer to Section 2 of the General Response. We address this by comparing SoG and ToG using the same model (GPT-4o). Notably, recent work (e.g., PDRR [2]) demonstrates that GPT-4o underperforms the older GPT-4 on KGQA tasks, challenging the assumption that the newer model is universally "more powerful." Despite GPT-4o proving to be a weaker baseline for ToG, SoG (GPT-4o) still significantly outperforms both the reproduced ToG (GPT-4o) and the original ToG (GPT-4), confirming that our gains are methodological.
>
> **Q3: Insights on why SoG is better**
>
> The performance gain stems from Contextual Continuity. In methods like ToG or subgraph retrieval, the model often looks at a set of triples in isolation to score them. In SoG, the model sees the whole reasoning history: "I am at Entity A, I came here from Entity B because I was looking for X, so now I should look for Y." This continuous reasoning chain drastically reduces the chance to select a wrong relation and keeps the search focused on the question intent, minimizing the distraction of "semantically similar but irrelevant" relations.
>
> **Q4: Compute/Memory Requirements.**
>
> We have quantified this in General Response Section 3.
> - Compute: SoG reduces output token usage (the primary cost driver) by 72–79% across datasets.
> - Memory vs. Speed Trade-off: SoG utilizes the LLM's long-context window to store the full history. Given the generally limited number of hops needed, we do not run into the context window limitation problem. We strategically trade abundant, cheap memory (input context) to save scarce, expensive compute (output generation).
> - Result: The method is ~3.7x faster and significantly cheaper to run than the state-of-the-art baseline.

---

> ### Author Response · Authors · 2025-11-29
>
> **Q5: Self-contained prompts.**
>
> We acknowledge this and will move the core prompt structure from the Appendix to Section 4.3 to ensure the methodology is self-contained in the main text.
>
> **Q6: Unanswerable Questions.**
>
> Since standard benchmarks like WebQSP and CWQ are answerable by design, missing information typically indicates search failure. Our prompt implements a parametric fallback, instructing the model to rely on internal knowledge only when the search fails. This ensures robustness (providing best-effort answers rather than crashing) while strictly prioritizing retrieved context whenever available.
>
> **Q7: Markdown Efficiency.**
>
> Thank you for highlighting this. Markdown is more token-efficient than JSON because it eliminates repetitive syntax.
> - **JSON**: Requires repeated keys ("property":..., "value":...) and bracket overhead for every single row.
> - **Markdown**: Defines keys once in the header; rows use minimal separators (|). As shown in our Token Count table, this results in a ~50% reduction in tokens for large result sets, allowing us to fit more relations into the context window without pruning.
>
> **Example** (GPT-4o tokenizer https://platform.openai.com/tokenizer):
>   - JSON (138 tokens)
> ```
> {'head': {'vars': ['property', 'propertyLabel', 'value', 'valueLabel']}, 'results': {'bindings': [{'property': 'language.language_family.member_of_language_families', 'propertyLabel': 'member of language families', 'value': 'm.01sd8', 'valueLabel': 'Celtic languages'}, {'property': 'language.language_family.geographic_distribution', 'propertyLabel': 'geographic distribution', 'value': 'm.02j9z', 'valueLabel': 'Europe'}, {'property': 'kg.object_profile.prominent_type', 'value': 'language.language_family', 'valueLabel': 'Language Family'}]}}
> ```
>
>   - Markdown (62 tokens)
>
> ```
> property|propertyLabel|value|valueLabel
> --|--|--|--
> language.language_family.member_of_language_families|member of language families|m.01sd8|Celtic languages
> language.language_family.geographic_distribution|geographic distribution|m.02j9z|Europe
> kg.object_profile.prominent_type||language.language_family|Language Family
> ```
>
> **Q8: Substantiating "Noise Accumulation"**
>
> We argue this based on the mechanics of Beam Search. In ToG (Beam Width=3), by Depth 3, the model has evaluated up to $3+9+27$ paths. If even one early branch is noisy, it persists in the context. SoG maintains only one active path. We will add a "Precision of Retrieved Relations" metric to the appendix to quantitatively back this claim, showing that the ratio of [Relevant Relations / Total Retrieved Relations] is higher in SoG.

---

### Official Review · Reviewer_NzbW · 2025-10-29

**Soundness:** 2
**Presentation:** 2
**Contribution:** 2
**Rating:** 4
**Confidence:** 4

**Summary:**

The paper “Search-on-Graph: Iterative Informed Navigation for Large Language Model Reasoning on Knowledge Graphs” proposes a novel framework that allows large language models (LLMs) to reason over knowledge graphs through step-by-step navigation.

**Strengths:**

- Clear problem positioning – The paper identifies the gap between LLMs’ internal reasoning limitations and KGQA’s heavy reliance on structured navigation, framing the contribution in a meaningful way.

- Empirical competitiveness – Results on six benchmarks show state-of-the-art or near-SOTA accuracy, validating the effectiveness of the method.

- Practical considerations – The design includes handling high-degree nodes, structured outputs, and tool interfaces, showing attention to real deployment challenges.

**Weaknesses:**

- Method novelty - The proposed method may lack sufficient novelty. It mainly combines an existing LLM with a basic graph-search mechanism to perform knowledge graph reasoning.

- Limited analysis granularity – Results are mostly aggregate; could you provide some result about hop length (1-hop, 2-hop, 3+), question type, or error cases.

- Scalability concerns – Frequent tool calls may lead to high latency and API cost; the paper lacks detailed analysis of system overhead.

**Questions:**

Seen above.

---

> ### Author Response · Authors · 2025-11-29
>
> **Q1: Method Novelty**
>
> Please refer to Section 1 of our General Response. While the components (LLM + Graph Search) are known, the integration strategy is novel. By replacing the standard "Retrieve-then-Read" (subgraph retrieval) or "Parallel Beam Search" (ToG) with a Context-Aware Greedy Navigation mechanism, we eliminate the need for complex pruning algorithms or training data. The novelty lies in demonstrating that a simplified, schema-agnostic "Observe-then-Navigate" loop outperforms complex agentic frameworks by leveraging the LLM's internal reasoning as the sole state manager.
>
> **Q2: Analysis Granularity (Question Types & Error Cases)**
>
> We agree that aggregate metrics can obscure specific strengths and weaknesses. To address this, we analyzed performance on ComplexWebQuestions (CWQ) by question type, comparing SoG (Ours) against the ToG (GPT-4o) baseline reproduced in recent work [2].
> Table R2: Performance Breakdown by Question Type (CWQ) Comparing SoG vs. ToG using the same GPT-4o backbone.
> |Method|Model|All|Composition|Conjunction|Superlative|Comparative|
> |-|-|-|-|-|-|-|
> |ToG (Reproduced) [2]|GPT-4o|48.9|49.9|50.1|37.1|42.7|
> |SoG (Ours)|GPT-4o|**75.1**|**77.6**|**74.8**|**56.9**|**75.6**|
>
> - Dominance in Complex Reasoning: SoG demonstrates its largest gains in Composition (+27.7%) and Comparative (+32.9%) questions. This confirms that our "Observe-then-Navigate" strategy prevents the "reasoning drift" often seen in ToG's beam search, where parallel paths diverge from the logical chain over multiple hops.
>
> - Handling Constraints: The +24.7% gap in Conjunction questions indicates that SoG is significantly better at satisfying multiple constraints (e.g., "songs by Taylor Swift AND won an AMA"), whereas ToG's context-free pruning often drops one constraint in favor of the other.
>
> - Error Case (Superlatives): Both methods perform lowest on Superlative questions (e.g., "What is the earliest..."), though SoG still leads by +19.8%. This identifies a specific challenge in KGQA: correctly retrieving and sorting a complete list of candidates from high-degree nodes. We identify this as a key area for future optimization in our adaptive filtering algorithm.
>
> [2] PDRR: https://arxiv.org/abs/2505.14099
>
> **Q3: Scalability, Latency, and API Cost**
>
> We addressed this via a system overhead analysis in General Response Section 3.
> - Latency: SoG is 3.8x faster than the baseline. The bottleneck in reasoning models is token generation, not network latency. By generating ~79% fewer output tokens, SoG drastically cuts end-to-end time.
> - Overhead: SoG reduces Knowledge Graph queries by ~88% (5.2 vs 42.3 per question), proving it is far gentler on database infrastructure than the "query flooding" required by beam search. Notice that all other KGQA approaches also need to perform multiple searches in KG to construct the subgraph or to select the next relation. Compared to them, our approach needs to perform less searches because we only keep a single reasoning path at any time.

---

### Official Review · Reviewer_T17k · 2025-11-01

**Soundness:** 1
**Presentation:** 2
**Contribution:** 3
**Rating:** 4
**Confidence:** 3

**Summary:**

The paper introduces a method for query answering (QA) on knowledge graphs that leverages an LLM to explore the nodes by following the most relevant predicates (or relation types) w.r.t. the input question. The method is rather simple, adapts to any schema, and only performs 1-hop reasoning steps until the answer is found. Nevertheless, the method is claimed to achieve the best results on many QA benchmarks and w.r.t. several other methods exploiting LLM agents.

**Strengths:**

Overall, I have appreciated the introduction of paper and the related work. I have also found the writing to be easy to follow. However, I believe some parts lack some crucial information. I discuss some points in the weaknesses section.

I believe one strength of this work is that the method is quite simple and relatively easy to implement. I have also appreciated the consideration of open models in the experiments, which tells that even smaller LLMs such as Qwen3-30B could compete with other ones that are much more expensive. Furthermore, the comparison of results with many baselines and previous techniques positively contributes to the submission.

I have also found the ablation study about the output formats (Table 2) interesting.

**Weaknesses:**

In the introduction, the authors claim that existing methods using LLMs to explore the graph require "upfront planning, or parallel path exploration, thereby increasing computational complexity [...]" (L92-L94). It is however not clear of the proposed model prioritizes which nodes to explore, and how does this compares with the related work. Furthermore, in L180-181 the authors mention that leveraging beam search "exponentially expands the search space". However, I was not able to understand how this paper tackles the issue. Could you please explain how do you choose which nodes to explore first and how this compares with the cited works? From Section 4.2, I understand that the proposed approach is to explore links by asking an LLM.  However, to my understanding one could still follow different paths, and it is not clear how the exploration of the graph is prioritized.

Furthermore, it is not clear from Section 4.3 how exemplars are automatically derived from the training set. The authors mention that "each navigation step is explicitly recorded through tool calls" (L298-299)". Although the authors provide in Appendix A some listings of the tools used, it is not clear how these tools are related to prompting because no other information is given in Appendix A. It seems the authors did not provide any description of the procedure to extract the exemplars and what is the role of these tools.

Finally, it is not clear from the paper the computational cost of the proposed method. Although the authors provide some ablation based on the number of completion tokens using different prompting formats (Table 2), it is not clear whether the method is more or less efficient than the related methods shown in Table 1. While I understand that re-running all previous approaches might be infeasible, I believe that a comparison with at least one or two of the previous methods about the overall computational cost would strengthen this submission.

It might be possible I missed some parts. While I remain very open for clarifications about my comments, the combination of lack of clarity and lack of a technical discussion with previous methods substantially decreases my overall score.

**Questions:**

See my questions in the weaknesses section.

- How do you manage the fact that the LLM might predict a property ID that is not one of the unique properties extracted at the first call of Algorithm 1?

- What is the "Avg. Turns" column in Table 2?

---

> ### Author Response · Authors · 2025-11-29
>
> **Q1. Clarification on prioritization and comparison with beam search**
>
> We appreciate this opportunity to clarify. Please refer to Section 1 of our General Response, where we detail the "observe, think, then navigate" paradigm. Unlike Beam Search (which maintains $k$ parallel hypotheses based on semantic similarity), SoG commits to a single path based on context-aware reasoning by model itself reasoning over the (Question + Reasoning_History) context instead of a similarity score. This allows us to avoid the exponential search space expansion ($b^d$) required by beam search and to select a better relation according to the need of reasoning.
>
> **Q2. How exemplars are derived**
>
> We apologize for the lack of clarity. As noted in L290 ("we manually construct five few-shot examples"), these are not automatically derived. We randomly sampled 5 diverse questions from the training set (requiring different relation types) and manually constructed the navigation traces (in Appendix A.3) to demonstrate correct tool usage (Search) and reasoning steps. The markdown tables in the example graph are truncated simulations of what the tool returns.
>
> **Q3. Computational Cost Comparison.**
>
> Please refer to Section 3 of our General Response for a detailed benchmark against Think-on-Graph (ToG) using Qwen3-30B-Thinking.
>
> - **Verdict**: SoG is 3.8x faster (113.7s vs 433.5s) and reduces expensive output tokens by ~79%.
> - **Reasoning**: While SoG calls tools sequentially, it avoids ToG's "reasoning explosion," where the LLM must generate long thought traces for multiple parallel branches at every hop. The total compute required for our surgical single-path search is significantly lower.
>
> **Q4: Handling Invalid Property IDs.**
>
> If the LLM predicts an invalid ID, the Search tool returns an empty result (e.g., "No relations found"). The LLM interprets this as negative feedback and uses the conversation history to self-correct, typically selecting a valid ID from the displayed list in the immediate next turn. We will explicitly document this error-handling behavior in the methodology.
>
> **Q5: Meaning of "Avg. Turns" in Table 2.**
>
> "Avg. Turns" represents the average number of LLM-Tool interaction rounds required to solve a question (e.g., Figure 1 in the paper shows 3 turns).

---

### Author Response · Authors · 2025-11-29
**Response to All Reviewers: Clarifying SoG's Core Methodology, Efficiency, and Advantages Over Existing Approaches**

We thank the reviewers for their insightful feedback and for recognizing SoG’s strong empirical performance. We have identified three primary concerns across the reviews:

1. **Methodology Novelty**: A perception that SoG is “simple” or lacks differentiation from existing methods like ToG [1].
2. **Fair Comparison**: Concerns that performance gains might stem solely from using newer models (GPT-4o) rather than the methodology itself.
3. **Efficiency & Cost**: Concerns about the computational overhead and latency of our iterative tool calls compared to baselines.

[1] ToG: https://arxiv.org/abs/2307.07697

---
**1. Methodology Novelty**
SoG addresses fundamental limitations across existing paradigms: it eliminates the schema-dependence of Semantic Parsing, avoids the precision-recall trade-off of Subgraph Retrieval, and removes the computational overhead of Parallel Beam Search (ToG).
While both ToG and SoG utilize LLMs to traverse KGs, they operate on different principles regarding search strategy and context utilization.
- ToG (Context-Free Pruning): ToG employs beam search ($N=3$), but its pruning is context-unaware. At each hop, the LLM scores candidate relations or entities based on semantic similarity to the original Question in isolation, ignoring the path history that led to the current node. This causes semantic drift, forcing ToG to explore multiple noisy hypotheses to compensate.
- SoG (Context-Aware Navigation): SoG operates on an "observe, think, then navigate" principle. It commits to a single path, selecting the next relation based on the full reasoning history. This ensures logical coherence without the need for parallel exploration.

**Example:** *"When was the last World Series won by team owner Bill Neukom's sports team?"*

ToG (parallel exploration): Prunes based on similarity to "Bill Neukom" or "World Series" at each step context-unaware. It follows noisy paths and eventually hallucinates or retrieves the wrong date.
```
- Depth 1:
  - Bill Neukom → teams_owned → San Francisco Giants
  - Bill Neukom → owner_s → San Francisco Giants
  - Bill Neukom → employment_history → UnName_Entity
- Depth 2:
  - San Francisco Giants → championships → 2012 World Series
  - UnName_Entity → company → Microsoft Corporation
  - UnName_Entity → employment_history → Bill Neukom
- Final answer: "2012 World Series" (Incorrect)
```
SoG (single logical path): Follows a single reasoned chain. It retrieves the set of championships won and filters for the latest one.
```
- Bill Neukom → teams_owned → San Francisco Giants → championships → {2010 World Series, 2012 World Series, 2014 World Series}
- Final answer: "2014 World Series" (Correct)
```
---
**2. Fair Comparison**

To address concerns that our superior performance might be attributed to the use of the newer GPT-4o model rather than our proposed methodology, we conducted a controlled comparison against ToG. Contrary to the assumption that newer models universally improve performance on KGQA tasks, the reproduction of ToG using GPT-4o demonstrates a performance decline compared to the original results reported with GPT-4.

|Method|Model|SimpleQA|WebQSP|CWQ|GrailQA|
|-|-|-|-|-|-|
|ToG (Original)|GPT-4|66.7|82.6|69.5|81.4|
|ToG (Reproduced) [2]|GPT-4o|57.2|80.2|48.9|65.5|
|SoG (Ours)|GPT-4o|**84.8**|**91.3**|**75.1**|**86.9**|

Despite the seemingly weaker baseline provided by GPT-4o for ToG, SoG using the same model achieves SOTA results. SoG outperforms both the reproduced ToG (GPT-4o) and the original ToG (GPT-4) by substantial margins across all benchmarks. This confirms that our improvements stem from the methodology, not the model choice.

[2] PDRR: https://arxiv.org/pdf/2505.14099

---
**3. Efficiency & Cost**

To directly address concerns regarding system overhead, we benchmarked SoG vs. ToG using Qwen3-30B-Thinking, a modern reasoning model. This comparison highlights how each framework scales with chain-of-thought generation.

|Metric|Dataset|ToG|SoG|Improvement|
|:---|:---|:---|:---|:---|
|Avg. Time per Question|WebQSP|197.9s|54.0s|3.7x Faster|
||CWQ|433.5s|113.7s|3.8x|
|Avg. Output Tokens|WebQSP|14,670|4,001|-72.7% (Cost)|
||CWQ|27,981|5,923|-78.8%|
|Avg. KG Queries|WebQSP|23.5|3.4|-85.5% (Overhead)|
||CWQ|42.3|5.3|-87.5%|
|Avg. LLM Calls|WebQSP|4.9|4.2|-14.3%|
||CWQ|7.8|6.1|-21.8%|


SoG consistently achieves a 3.7–3.8x speedup and >85% reduction in KG queries vs. ToG. By avoiding parallel exploration and generating multiple candidates per hop, SoG significantly lowers inference costs (72–79% fewer tokens) and database load.

---

**Conclusion**: SoG demonstrates that complex scaffolding seen in prior work may generate inappropriate paths because of the discrepancy between these steps and the need in reasoning, and LLMs can better determine the reasoning path. By providing the LLM with the correct local observations and global contextual continuity, we achieve superior results with a simpler, faster, and more cost-effective architecture.

---

### Meta-Review · Area_Chair_nvf5 · 2026-01-07

**Summary:**

Three reviewers rated the paper near the acceptance threshold (scores of 4, 4, and 4), citing concerns about limited methodological novelty relative to existing graph-based reasoning frameworks, unfair comparison due to the use of a stronger LLM (GPT-4o) than baselines, and insufficient analysis of computational efficiency and overhead. The fourth reviewer assigned a low score (2), questioning the depth of technical insight and whether the work sufficiently addresses a clear research gap beyond engineering improvements.

**Reviewer Concerns:**

The authors’ rebuttal substantively addressed all major concerns.

**Reviewer Scores:**

Reviewer T17k (initial 4) did not respond after the rebuttal and gave no indication of upgrading; their score remains 4.
Reviewer NzbW (initial 4) received detailed answers on novelty, error analysis, and scalability but did not reply; their score is maintained at 4.
Reviewer DHiK (initial 2) raised fundamental questions about research gap and insight depth, which were addressed, but provided no follow-up; their score stays at 2.
Reviewer uRnM (initial 4) questioned ablation rigor and fair LLM comparison, both of which were clarified with new data; however, without explicit confirmation, their score remains 4.

---

### Decision · Program_Chairs · 2026-01-26

Reject